# Endometriosis in Menopause—Renewed Attention on a Controversial Disease

**DOI:** 10.3390/diagnostics10030134

**Published:** 2020-02-29

**Authors:** Cristina Secosan, Ligia Balulescu, Simona Brasoveanu, Oana Balint, Paul Pirtea, Grigoraș Dorin, Laurentiu Pirtea

**Affiliations:** 1Department of Obstetrics and Gynecology, University of Medicine and Pharmacy “Victor Babeş”, 300041 Timişoara, Romania; cristina.secosan@gmail.com (C.S.); brasoveanu_simona@yahoo.com (S.B.); oana.balint@gmail.com (O.B.); grigorasdorin@ymail.com (G.D.); laurentiupirtea@gmail.com (L.P.); 2Department of Ob Gyn and Reproductive Medicine, Hopital Foch—Faculté de Medicine Paris Ouest (UVSQ), 92151 Suresnes, France; paulpirtea@gmail.com

**Keywords:** endometriosis, menopause, diagnosis, management, malignancy

## Abstract

Endometriosis, an estrogen-dependent inflammatory disease characterized by the ectopic presence of endometrial tissue, has been the topic of renewed research and debate in recent years. The paradigm shift from the belief that endometriosis only affects women of reproductive age has drawn attention to endometriosis in both premenarchal and postmenopausal patients. There is still scarce information in literature regarding postmenopausal endometriosis, the mostly studied and reported being the prevalence in postmenopausal women. Yet, other important issues also need to be addressed concerning diagnosis, pathophysiology, and management. We aimed at summarizing the currently available data in literature in order to provide a concise and precise update regarding information available on postmenopausal endometriosis.

## 1. Introduction

The concept that endometriosis is a disease that only affects women of reproductive age has prevailed since 1942, when the first case of endometriosis in a postmenopausal patient was reported by Edgar Haydon [1]. 

In spite of this early report, endometriosis has also been described in premenarchal patients and is a common occurrence in adolescents [2,3,4]. 

The recurrence of endometriosis lesions in patients with a prior diagnosis of endometriosis during the premenopausal period or the de novo appearance of endometriosis in postmenopausal patients with no prior history of endometriosis-related complaints has been however well documented in numerous case series, case reports, and retrospective studies [5,6,7,8,9]. 

The management of endometriosis in postmenopause and hormone replacement therapy (HRT) in patients with a history of endometriosis remains controversial.

## 2. Prevalence

The incidence of postmenopausal endometriosis reported in literature is of approximately 2–5%. It commonly represents a side effect of HRT, rarely occurring in patients without a history of HRT or Tamoxifen treatment [10]. In a few cases, postmenopausal endometriosis has been described in women who had no history of endometriosis on imaging or surgery prior to menopause [11].

In order to evaluate the prevalence of postmenopausal endometriosis, Haas et al. performed a retrospective epidemiological study on 42,079 women admitted for surgical treatment with histologically confirmed endometriosis. Patients were sorted in 5 years age groups and also in premenopausal, perimenopausal, and postmenopausal subgroups. The results showed that 33,814 patients (80.36%) were in the premenopausal group (age 0–45 years), with 23 patients (0.05%) being younger than 15 years; of the remaining patients, 7191 (17.09%) were in the perimenopausal (45–55 years), and 1074 patients (2.55%) in the postmenopausal group, respectively [6].

## 3. Pathophysiology

Endometriosis is an estrogen-dependent inflammatory disease characterized by the presence of ectopic endometrial tissue. The pathogenesis of endometriosis remains enigmatic [12]. 

Postmenopausal endometriosis is considered to have an even more complex pathophysiology than premenopausal endometriosis. It is still unclear whether this represents a recurrence or continuation of a previous disease or a de novo condition. Excess estrogen, in general, represents a promoting factor for endometriosis. The arrest of estrogen production at the level of the ovaries at the time of menopause is counterbalanced by peripheral estrogen production from conversion of androgens (especially in the adipose tissue and skin). The leading estrogen found in these patients is estrone.

An attractive theory regarding the pathogenic mechanism of postmenopausal endometriosis involves the “estrogen threshold”, i.e., when a certain estrogen level is reached or surpassed in postmenopausal patients it activates undetected or “transient” foci of endometriosis.

In addition to the peripheral estrogen production, a high circulating level of estrogen may be of external source, especially in the form of phytoestrogens and HRT. Phytoestrogens appear to exert estrogenic effects on the uterus, breast, and pituitary and could also support the growth of endometriotic lesions [13,14,15].

Despite the fact that postmenopausal endometriosis has the same immunochemical profile as premenopausal endometriosis and has the potential to reactivate under estrogen stimulation, endometriosis lesions in the postmenopausal period seem to be less common, less extensive, and less active in most cases [16].

## 4. Symptomatology

The clinical presentation of endometriosis in menopausal patients is unspecific, such as pelvic pain, ovarian cysts, or intestinal symptoms. Given the age of the patients, they are often suspected of a neoplastic process. As a general consideration, all postmenopausal patients should be evaluated for malignancy if a new suspicious structure is found on ultrasound examination.

In menopausal women with a history of endometriosis, the drop in estrogen levels after menopause relieves the endometriosis-related symptoms but generates specific menopausal ones, such as mood swings, hot flushes, vaginal atrophy, and night sweats [5,17]. The clinical grim reality is that the severity of the disease is not necessarily reflected in the degree of discomfort. Commonly, the complaints of pelvic pain underestimate the disease’s severity in both premenopausal and postmenopausal endometriosis.

## 5. Diagnosis

Despite intensive research conducted in the last decades, endometriosis remains a disease with a delayed diagnosis, especially in older patients. This results from the lack of noninvasive tools available for early stage diagnosis. For many years, there has been a long-standing myth that endometriosis is a disease that affects only adult women of reproductive age. However, in recent years, focus has turned to the diagnosis of endometriosis in postmenopausal patients, given that the onset of pain can start after the onset of menopause, with reports of endometriosis occurring even in 80-year-old patients [1,5].

The ovaries are the most common location of endometriotic lesions in postmenopausal patients (79.2% of cases) [18].

Distinction between endometriosis lesions and cancer is complicated by the fact that some of the risk factors are similar, such as low parity rate, infertility, late childbearing age, and a short duration of oral contraceptive use [19].

Currently, laparoscopy and biopsy for histological confirmation of suspicious lesions is the gold standard for diagnosis of endometriosis, irrespective of age. Laparoscopy, the standard technique for inspecting the pelvis, can provide simultaneous diagnosis and treatment of lesions. Additional tools are needed for a noninvasive diagnostic and classification. To this date, no serum marker or test is available for reliably diagnosing endometriosis [20,21]. Regarding imaging investigations, MRI and ultrasound are important, but findings are more difficult to interpret in menopausal patients than in younger patients due to the higher suspicion for neoplastic lesions and the polymorphic aspect of endometriosis.

### 5.1. Clinical Examination

The patient’s medical history, clinical examination, or preoperative symptoms have a limited role in determining the extent of endometriosis lesions as there is no direct relationship between symptoms and the anatomic-surgical characteristics of endometriotic lesions [22]. Also, there is usually a discrepancy between the severity of symptoms and the extent of lesions with many patients whose severe lesions remain asymptomatic. This is an important factor contributing to a delay of approximately 6 to 8 years from onset of symptoms to diagnosis in premenopausal and postmenopausal patients alike [23].

Pelvic vaginal and rectal examination is useful in identifying endometriosis nodules in the lower posterior compartment, but clinical examination may be normal in many patients with deep infiltrating endometriosis [23].

### 5.2. Imaging

While diagnostic laparoscopy remains the gold standard, it is often not the first line of diagnosis any more, as noninvasive testing for early diagnosis and progression of endometriosis is being preferred [24]. Yet, no imaging method can definitively confirm the diagnosis of endometriosis, being notably inconclusive in case of peritoneal implants [25].

Deep infiltrating endometriosis (DIE) can be investigated through several imaging techniques, including transvaginal sonography (TVS), magnetic resonance imaging (MRI), computerized tomography, rectal endoscopic sonography, and three-dimensional (3D) ultrasound [23].

TVS has gained interest in recent years and is starting to be recommended as the first-line investigation technique in endometriosis because it allows extensive exploration of the pelvis, is widely available, cost efficient, and well tolerated [26,27,28]. 

TVS has the benefit of a lack of exposure to radiation and is the main method for the evaluation of adnexal masses, but remains limited for the diagnosis of other kinds of endometriosis. Endometriomas have distinct characteristics on ultrasound: unilocular cysts, most often of homogenous “ground glass” appearance. The identification of an endometrioma should alert the clinician to the possibility of moderate-to-advanced stage disease. An important exception is postmenopausal women, in whom ovarian cysts with a “ground glass” appearance are associated with a 44% risk of malignancy. In addition, TVS may have a role in assessing disease involving the bladder and rectum [29].

Computed tomography (CT) plays a major role in the diagnosis of bowel endometriosis in the presence of colon distension. Genitourinary tract involvement should be taken into consideration in case of hydronephrosis or hydroureter diagnosed on CT scan, especially in patients with a history of chronic pelvic pain or in patients with a history of endometriosis. Radiation exposure should be taken into consideration [29].

MRI is a noninvasive diagnostic method of DIE that offers the possibility to fully investigate the pelvic cavity with a high accuracy, but increased costs [30]. Nevertheless, MRI has limited indication in the diagnosis of endometriosis. It can confirm the diagnosis of endometrioma in the presence of an adnexal mass when TVS is uncertain. MRI can also be used as an investigation method when involvement of the ureter is suspected, and may be beneficial in the evaluation of anatomy when expanded pelvic adhesions are suspected [29].

Sonovaginography using saline solution (saline contrast sonovaginography (SCSV)) or gel infusion sonovaginography is a new diagnostic method in DIE. First described by Dessole et al., it consists of TVS combined with the introduction of saline solution or gel infusion into the vagina, which offers the benefit of a more complete view of the vaginal walls and fornix, pouch Douglas, uterosacral ligaments, and rectovaginal septum [22]. The data available in literature is limited, with only a few reports from Brazil, Italy, Romania, and Australia, but the methods seems beneficial in the diagnosis of posterior deep infiltrating endometriosis. Up to date, no studies have reported its use in postmenopausal patients [22,31,32,33].

The role of double-contrast barium enema (DCBE) in the evaluation of rectovaginal endometriosis is controversial. Some studies have reported high accuracy in predicting the need for intestinal surgery in endometriosis cases. The superiority of DCBE over rectal ultrasound or MRI is not well established, the results reported in literature being scarce and contradictory. However, certain studies have demonstrated a lower sensitivity of DCBE for rectovaginal disease. DCBE does not allow the examination of the entire intestinal wall thickness and does not provide information regarding the depth of infiltration, but may provide useful information for preoperative planning when severe disease is suspected [29].

### 5.3. Biomarkers

To this date, no specific markers for the diagnosis of endometriosis have been identified. A change in levels of proteins, microRNAs, and other markers corresponding to a disease state could be the basis for identifying novel biomarkers. Endometriosis patients often show modified ranges of CA-125 (Cancer Antigen 125), cytokines, angiogenic and growth factors compared with normal women, but all of these biomarkers are frequently encountered in various other pathologies and are not specific enough for diagnosing endometriosis. A combination of biomarkers may improve the sensitivity and specificity over single biomarker measurements. Moreover, stem cell, proteomic, and genomic studies could contribute to the development of new high-sensitivity biomarkers in the diagnosis of endometriosis in the future [24].

Many authors have studied the role of biomarkers for diagnosis of endometriosis and concluded that, to date, endometrial tissue, menstrual or uterine fluids, and immunologic markers in blood or urine are not recommended for clinical use for diagnosis of endometriosis [24].

Regarding the differential diagnosis between endometriomas and malignant ovarian tumors in postmenopausal patients, we have not found any information in the literature that supports the use of any novel tests, such as OVA1 (Ovarian Malignancy Algorithm), ROMA (risk of ovarian malignancy algorithm), circulating miRs, etc. Despite the potential clinical utility of these biomarkers in the diagnosis of malignant ovarian tumors in premenopausal patients, the costs implied, the lack of easy availability, and the decreased incidence of endometriomas in older patients make the usefulness of novel biomarkers difficult to assess [34,35,36]. 

### 5.4. Minimally Invasive Surgery: Laparoscopy and Robot Assistance

Because of the lack of specific and efficient noninvasive tests for endometriosis, there is often a significant delay in diagnosis of this disease, especially in older patients. The gold standard for the diagnosis of endometriosis remains visual inspection by laparoscopy, preferably with histological confirmation. A positive histological examination confirms the diagnosis, but negative histology does not exclude it, in the presence of pathognomonic lesions [23].

Whether histology should be obtained if peritoneal disease alone is present is controversial: a visual inspection of the pelvis should be enough, but histological confirmation of at least one lesion is ideal. In some cases, histology should be obtained to identify endometriosis and to exclude malignant disease. For example, in ovarian endometriomas (>3 cm in diameter) and in deeply infiltrating disease, a histological confirmation to exclude a rare instance of malignancy is necessary [37].

Laparoscopy is used for the diagnosis and treatment of DIE and serves to eradicate all visible endometriosis implants, especially in postmenopausal patients due to the risk of malignant transformation. Several studies have shown a significant improvement of symptoms and a decreased risk of malignancy in postmenopausal women after complete resection of all visible lesions. Precise preoperative imaging may help guide surgical therapeutic approaches and aid to obtain the best postoperative results [23]. In the last years, the da Vinci surgical system started to be used in the diagnosis and treatment of endometriosis. Three-dimensional (3D) vision offers the advantage of improved depth perception and accuracy in the performance of robotic surgery, particularly for complex surgical tasks such as identifying suspected implants. However, the robotic platform has the distinct disadvantage of offering only a unidirectional view within the abdominal cavity. Authors recommend for the first instance to undertake a diagnostic laparoscopy to exclude a suspected lesion of endometriosis in the upper abdomen, liver, diaphragm, and appendix before using the da Vinci robotic system in the pelvis. Another disadvantage is the loss of haptic feedback to identify fibrotic lesions which are characteristic of deeply infiltrating disease. However, the da Vinci robot may offer improved ease by avoiding hand and more instinctual movement of the wristed instruments in the treatment of endometriosis. The cost related to the procedure also make it unavailable at a large scale [29].

## 6. Management

### 6.1. The Impact of Hormone Replacement Therapy in Women with a History of Endometriosis 

The recently published guidelines on menopause management have no statements of endometriosis symptoms [38]. The use of HRT raises concerns about disease reactivation and recurrence of pain and need for surgical treatment, and even malignant transformation of residual endometriosis. The risk of recurrence with HRT is considered to be linked to residual disease after surgery. The data regarding hormone therapy regimens is scarce. Continuous combined estrogen–progesterone treatment or tibolone, in patients with or without hysterectomy, is considered to carry a lower risk of disease recurrence, compared with estrogen-only regimens, but larger studies are required in order to prove the safety and efficacy. Management of potential recurrence is best monitored by awareness of the possibility of symptom recurrence. Patients with contraindication or who refuse hormonal treatment should be offered alternative pharmacological treatment for menopausal symptoms and for skeletal protection, if indicated. Herbal products should be avoided as some may contain estrogenic compounds and their efficacy is uncertain [39,40,41]. The risk of malignant transformation of endometriosis in women with a history of endometriosis who received HRT remains a matter of debate. Long-term follow-up studies are needed to evaluate the risk of an adverse outcome. Further studies are mandatory in order to determine the optimal management of menopause in women with endometriosis [15].

### 6.2. The Management of De Novo Endometriosis in Postmenopausal Patients and Pain Management

“De novo” endometriosis appears especially after unopposed estrogen therapy or obesity, which has an additional effect for increasing the risk of endometriosis development.

Postmenopausal women with symptomatic endometriosis should be managed surgically with removal of all visible endometriotic tissue because of the higher risk of recurrence and the risk of malignancy [41]. A similar approach is recommended by current ESHRE (European Society of Human Reproduction and Embryology) recommendations. Medical therapy can be used in case of pain recurrence after surgery or if surgery is contraindicated. Co-morbidities represent an additional risk to contraindicate surgery and include advanced age or pelvic adhesions from previous surgery [38,41]. Approximately 12% of all endometriosis cases will finally require a hysterectomy with or without oophorectomy [42,43]. To prevent recurrences, to restore bowel, urinary, or sexual function or to alleviate pain it is now recommended to remove all the implants [38].

Progesterone administration (oral or intrauterine system) has been proposed as a reliable alternative treatment in patients with contraindication for surgery, but, up to date, no extensive data is available and further studies are needed regarding progesterone use in postmenopausal endometriosis [44,45].

Aromatase inhibitors act by decreasing extra-ovarian estrogen production and by blocking the feed-forward stimulation loop between inflammation and aromatase within endometriosis lesions. Only six case reports of aromatase inhibitors administration in postmenopausal patients with a history of endometriosis have been published so far. In 1998, Kayama presented a 57-year-old patient who had presented with recurrent endometriosis with a painful vaginal polypoid mass. The use of Anastrozole reduced the volume of the vaginal mass. Other studies concluded that Letrozole has a similar efficacy to Anastrozole [46,47,48]. The most important risk of this treatment is osteoporosis and related fractures. Aromatase inhibitors impair bone mineral density and need to be associated with bisphosphonate therapy.

## 7. Tamoxifen and Postmenopausal Endometriosis

Tamoxifen represents a hormonal substitution therapy used in postmenopausal women with breast cancer. Tamoxifen has antiestrogenic effects on breast tissues but promotes endometriosis through unknown mechanisms. In 1993, the first case of tamoxifen-related endometriosis was reported in a woman who received tamoxifen for 2 years due to breast cancer [49]. In the next year, it was reported another case of operated breast cancer and adjuvant tamoxifen [50]. During the next years, many authors reported cases of ovarian and endometrioid carcinoma in the women who had used tamoxifen [51,52,53,54]. Considering that there is no significant statistical evidence, the relation between tamoxifen and malignant transformation may be coincidental [43].

## 8. Risk of Malignant Transformation

The possible transformation of endometriosis lesions into malignant lesions and their dissemination in the ovaries, bowel, and even lungs has been described. The risk of malignant transformation of endometrioma into an ovarian cancer is estimated at 2% or 3% [41,55], and may be higher in patients receiving estrogen therapy. Furthermore, patients with endometriosis have an increased risk of other malignancies, apart from ovarian cancer [41].

Differential diagnosis of benign from malignant tumors in postmenopausal women is difficult. We must take into account that some endometriosis lesions may have a similar appearance to malignant disease and can cause local and distant metastases and can infiltrate adjacent tissues and organs. Age is an important risk factor for many malignancies, thus it may be questioned whether the postmenopausal endometriosis increases the risk for malignancy [41].

In 1997, Brinton et al. showed that patients with endometriosis seem to have an increased overall cancer risk [56]. Some authors indicate an increased risk of ovarian cancer, calculated to be around 35% for clear cell carcinoma and 19% for endometrioid type carcinoma in women with endometriosis [57]. 

On the other hand, Somigliana et al. concluded that there is evidence to support that endometriosis should be considered a medical condition associated with a clinically relevant risk of any specific cancer [58].

Regarding the relationship between endometriosis and breast cancer, Bertelsen et al. published a study which followed around 115,000 Danish women over a period of 30 years. Authors concluded that the risk for breast cancer increased with age (<40 years) at diagnosis of endometriosis and it is around 0.97%. The increased risk associated with endometriosis among postmenopausal women may be due to common risk factors between postmenopausal endometriosis and breast cancer or an altered endogenous estrogen [59].

Because endometriosis and ovarian malignancy have some common risk factors, including low parity rate, infertility, late childbearing age, and short duration of oral contraceptive use, in clinical practice is very difficult to discriminate a benign from a malignant tumor in postmenopausal women [60].

In postmenopausal women who underwent surgery for endometriosis, hormonal therapy remains controversial. Unopposed estrogen stimulation is associated with an increased risk of endometrial cancer. Some studies show that exogenous estrogens are increasing the risk of malignancy transformation of endometriosis lesions. In a retrospective study which followed 31 women with cancer developing from endometriosis, Zanetta et al. concluded that prevalence of endometriosis associated with co-existing risk factors (obesity and unopposed estrogen therapy) represents a significant risk factor for the development of cancer in endometriotic lesions [61].

The indication for initiating hormone therapy in women with endometriosis must be carefully evaluated. In premenopausal women who underwent total hysterectomy and bilateral salpingo-oophorectomy due to endometriosis, the benefits of hormone therapy outweigh the risks. Postmenopausal hormone therapy may increase the risk of malignant transformation or recurrence of endometriosis [41,62]. More data are needed to confirm this.

## 9. Extrapelvic Endometriosis

Extrapelvic endometriosis is a rare clinical condition in postmenopausal women. It affects a slightly older population due to the fact that it takes several years for pelvic endometriosis to metastasize outside the pelvis. Statistical data regarding menopausal patients are limited. The most common location of extrapelvic endometriosis is the gastrointestinal tract, followed by the urinary system. Bladder and ureteral endometriosis are the most common sites for urinary tract involvement [63]. Regarding the gastrointestinal tract, the sigmoid colon is the most commonly involved, followed by the rectum, ileum, appendix, and caecum [64,65]. Extremely rare locations that have been reported include the gallbladder, the Meckel diverticulum, stomach, and endometriosis cysts of the pancreas and liver [63].

Flyckt et al. presented a 59-year-old woman with a periaortic mass with ureteral obstruction. A computed tomography was performed, and a surgical management was necessary to resect the mass. The pathology exam confirmed endometriosis invasion of the inferior vena cava [66].

### 9.1. Gastrointestinal Tract Endometriosis

In postmenopausal women with low estrogen levels, a vascular transport or metaplasia of intestinal tissue should be considered for the etiology of gastrointestinal tract endometriosis [65]. The intestinal involvement in endometriosis after menopause is a rare phenomenon and it has been described in literature only in case reports (Figure 1). 

Snyder et al. presented the case of a woman with iron-deficiency anemia, who underwent total hysterectomy with bilateral salpingo-oophorectomy. During the surgical procedure, an endometrial implant at the hepatic flexure was discovered, a rare location for endometriosis. The patient was treated with conjugated estrogen–bazedoxifen to antagonize the effects of estrogen. No evidence of lesion was found at colonoscopy after five months of therapy [11].

Popoutchi reported a rare case of postmenopausal intestinal endometriosis simulating a malignant lesion in a woman who previously underwent hysterectomy with bilateral salpingo-oophorectomy, with no hormone replacement treatment. She was treated by rectosigmoidectomy with colostomy [65].

It is difficult to diagnose bowel endometriosis by colonoscopy because most cases do not infiltrate beyond the serosa and very few infiltrate the mucosa [67]. Deep endometriosis is a very complicated disease to diagnose and treat, especially in older patients [43,68].

Jones et al. reported a case of a surgical menopause for deep rectovaginal endometriosis, with estrogen replacement therapy. A polyp was detected on colonoscopy and the biopsy confirmed a malignant transformation of endometriosis to adenocarcinoma [69].

### 9.2. Urinary Tract Endometriosis

Urinary tract endometriosis is an uncommon pathology and a silent cause of monolateral or bilateral kidney dysfunction. The diagnosis of urinary tract endometriosis is difficult since the disease is associated with nonspecific symptoms, regardless of the hormonal status [70] (Figure 2). 

A few case reports have been published regarding bladder endometriosis in postmenopausal women. Stewart reported a case of bladder endometriosis extending into the bowel in a postmenopausal woman. He concluded that it was due to reactivation of the endometrial implants under exogenous estrogen stimulation [71]. Also, a case of a 68-year-old postmenopausal woman, with no exogenous estrogen therapy, with an abnormal mass of the bladder that turned out to be an endometriosis lesion, was reported. This case suggests that endometriosis may persist even after years of a hormonally castrated state [72].

## 10. Conclusions

The paradigm shift from the belief that endometriosis only affects women of reproductive age has drawn attention to endometriosis in both premenarchal and postmenopausal patients. Despite its relatively low incidence, physicians should consider endometriosis in cases of unclear pelvic pain in postmenopausal patients, even if the patient has no prior history of endometriosis lesions. 

Postmenopausal endometriosis seems to expose the patient to a higher risk of malignant transformation. Due to the lack of high-quality studies, it remains unclear how to advise women with a history of endometriosis regarding the management of menopausal symptoms. The absolute risk of disease recurrence and malignant transformation cannot be quantified, and the impact of HRT use on these outcomes is not known. Multicenter randomized trials or large observational studies are urgently needed to inform clinicians and patients alike.

## Figures and Tables

**Figure 1 diagnostics-10-00134-f001:**
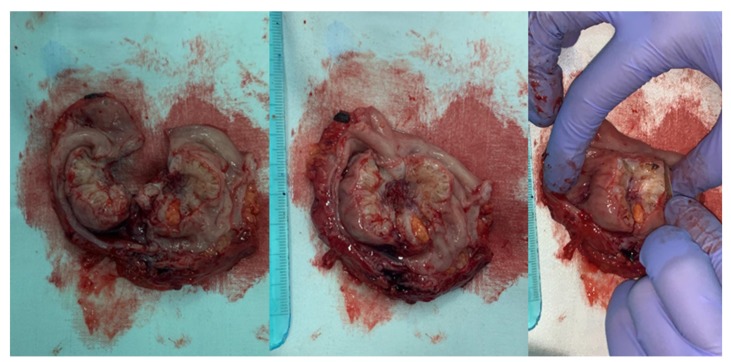
Sigmoid colon endometriosis—macroscopic aspect of the piece after laparoscopic resection using a circular stapler (personal collection, L. Pirtea).

**Figure 2 diagnostics-10-00134-f002:**
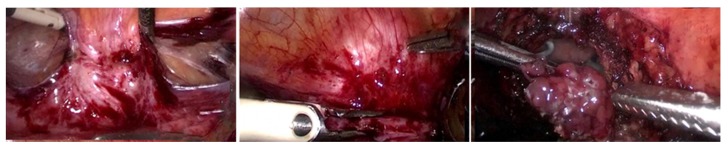
Urinary tract endometriosis—laparoscopic resection of a bladder endometriosis nodule infiltrating the bladder mucosa (personal collection, L. Pirtea).

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
