# Peer review of "Endometriosis in Menopause—Renewed Attention on a Controversial Disease"

_diagnostics, 2020, doi:10.3390/diagnostics10030134_

Round 1

Reviewer 1 Report

The paradigm shift from the belief that endometriosis affects only women of reproductive age is clinically important and essential for many obstetricians and gynecologists. As such, the content of this review is valuable to the journal. However, there are some parts of the review where it is difficult to understand whether the explanation is about general endometriosis or postmenopausal endometriosis. If these points are clarified, it will be worth to be published in this journal.

Author Response

Dear reviewer,

Thank you for taking the time to evaluate our manuscript.

We consider your comments very valuable and we have modified our manuscript according to them, as follows:

You suggested:  However, there are some parts of the review where it is difficult to understand whether the explanation is about general endometriosis or postmenopausal endometriosis. If these points are clarified, it will be worth to be published in this journal.

Response: We appreciate your interest in our manuscript. We agree and find this comment to be very accurate. We have made the requested modifications in the text.

We would like to thank you very much for your recommendations. We have found them to be extremely accurate and valuable, and therefore we consider that modifying the manuscript according to your specifications has increased its value.

Best regards,

Cristina Secosan

Reviewer 2 Report

General Comments:

Although the authors provide a general overview of endometriosis, their focus should be concentrated on post-menopausal patients (as stated in title). They need to organize their thoughts and their delivery to the readers about why this is an important topic they are writing about.

Are the authors attempting to diagnose patients/treat patients with known prior endometriosis or are they drawing attention to the fact that some patients without knowledge of prior disease may have endometriosis. If patients were unclear about prior diagnosis, were they questioned about menstrual symptoms? Do they have any of their own data that they wanted to present or is this strictly a review?

They need to have more up to date papers and check their references as some of them do not correspond to the citation.

Introduction:

-recurrence and de novo contradict each other

Pathophysiology:

-Excess estrogen in general is a factor that promotes endometriosis

Symptomatology:

-Patients should be evaluated for malignancy if new suspicious mass found on ultrasound.

-Newer tests such as OVA1 can be used to help establish an adnexal mass from a benign one.

Diagnosis:

-TVUS can determine if there are endometriomas, but otherwise is limited to what other kinds of endometriosis

-Pelvic pain and adnexal mass – always need to determine if the lesion is cancer or not.

-No imaging modality can definitively say that a patient has or does not have endometriosis

-Laparoscopy and Robotic Assisted Laparoscopy are both Laparoscopic techniques and have been in the area of surgical treatment for over 20 years.

Management

-Hormones such as OCPs, Lupron, or progestin only can help regulate menses and recurrence as shown by several studies.

-Have there been any studies that have looked at low dose progesterone only methods?

-Authors present a few case studies with post-menopausal patients and locations of where endometriosis is found.

-Authors do not present any new information to the readers. They do however present some good questions about how post-menopausal endometriosis should be treated.

Author Response

Dear reviewer,

Thank you for taking the time to evaluate our manuscript.

We consider your comments very valuable and we have modified our manuscript according to them, as follows:

General Comments:

You stated: Although the authors provide a general overview of endometriosis, their focus should be concentrated on post-menopausal patients (as stated in title). They need to organize their thoughts and their delivery to the readers about why this is an important topic they are writing about.

 Response: We have clarified in the text the possible misinterpretations regarding the subject and hope it is now more clearly understood that the focus of the manuscript is on postmenopausal endometriosis.

You asked: Are the authors attempting to diagnose patients/treat patients with known prior endometriosis or are they drawing attention to the fact that some patients without knowledge of prior disease may have endometriosis. If patients were unclear about prior diagnosis, were they questioned about menstrual symptoms? Do they have any of their own data that they wanted to present or is this strictly a review?

Response: Our manuscript is a review of literature, hence the data presented is not our own. We are taking into discussion the diagnosis and treatment of endometriosis in postmenopausal patients with a prior history on endometriosis, but also in patients with no endometriosis symptoms prior to menopause onset – de novo endometriosis (i.e. pain or menstrual symptoms). 

You stated: They need to have more up to date papers and check their references as some of them do not correspond to the citation.

Response:  We agree and have updated the reference list accordingly.

You stated : Introduction: -recurrence and de novo contradict each other

 Response: We agree the phrase may have been ambiguously formulated. We have rephrased the sentence and hope it is clearer to understand now.

You stated: Pathophysiology: -Excess estrogen in general is a factor that promotes endometriosis

Response: We agree and have added this statement in the text.

You stated: Symptomatology: -Patients should be evaluated for malignancy if new suspicious mass found on ultrasound.

Response: We agree with your statement and have added the information in the text.

You stated: -Newer tests such as OVA1 can be used to help establish an adnexal mass from a benign one.

 Response: We agree that novel biomarkers are useful in establishing an adnexal mass from a benign one, but we have not found any information in literature regarding their use in postmenopausal patients with endometriomas.

You stated: Diagnosis: -TVUS can determine if there are endometriomas, but otherwise is limited to what other kinds of endometriosis

Response: We agree with your statement and have modified in the text.

You stated: -Pelvic pain and adnexal mass – always need to determine if the lesion is cancer or not.

Response: We agree and have added the information in the previous section as suggested.

-No imaging modality can definitively say that a patient has or does not have endometriosis

Response: We strongly agree with your statement and have added the information in the text.

You stated: -Laparoscopy and Robotic Assisted Laparoscopy are both Laparoscopic techniques and have been in the area of surgical treatment for over 20 years.

 Response: We find you affirmation very valuable and pertinent and we agree that both laparoscopy and robot assisted laparoscopy represent similar minimally invasive techniques. We have modified the body of the text according to your suggestion.

You stated: Management -Hormones such as OCPs, Lupron, or progestin only can help regulate menses and recurrence as shown by several studies.

Response: We agree the hormonal treatment is beneficial in the treatment of endometriosis in premenopausal patients. Given our focus is mainly on postmenopausal patients, the use of hormonal therapy does not help regulate menses, but remains a valuable alternative in the management of patient with contraindications for surgical treatment.

 You asked: -Have there been any studies that have looked at low dose progesterone only methods?

Response: We have added the requested information regarding hormonal therapy by progesterone only in the text of the manuscript as suggested.

You stated: -Authors present a few case studies with post-menopausal patients and locations of where endometriosis is found. Authors do not present any new information to the readers. They do however present some good questions about how post-menopausal endometriosis should be treated.

Response: We value your appreciation regarding the questions depicted in our manuscript.

We would like to thank you very much for your recommendations. We have found them to be extremely accurate and valuable, and therefore we consider that modifying the manuscript according to your specifications has increased its value.

Best regards,

Cristina Secosan

Reviewer 3 Report

Comment 1: 

Recommendation of TVS as first line investigation by Authors should be based on some international guideline or consensus group. 

Provide more autheticated reference of above claim and replace Reference 23 with RCOG , ACOG or Euorpean guideline ,ESHRE etc reference 

Comment 2: SCSV is new method but Authors should mention whether its done or not in any part of Europe or world??  

Comment 3: correct spelling ''urinary sistem'' in the paragraph under heading of (9: extrapelvic endometriosis )

Author Response

Dear reviewer,

Thank you for taking the time to evaluate our manuscript.

We consider your comments very valuable and we have modified our manuscript according to them, as follows:

You stated: Comment 1: Recommendation of TVS as first line investigation by Authors should be based on some international guideline or consensus group. Provide more autheticated reference of above claim and replace Reference 23 with RCOG , ACOG or Euorpean guideline ,ESHRE etc reference 

Response: We appreciate your suggestion and have added the requested references in the text.

You stated: Comment 2: SCSV is new method but Authors should mention whether its done or not in any part of Europe or world??  

Response: We agree and we have added the requested information in the text of the manuscript.

You stated: Comment 3: correct spelling ''urinary sistem'' in the paragraph under heading of (9: extrapelvic endometriosis )

Response: Thank you for your vigilant remark. We have corrected in the text.

We would like to thank you very much for your recommendations. We have found them to be extremely accurate and valuable, and therefore we consider that modifying the manuscript according to your specifications has increased its value.

Best regards,

Cristina Secosan